# Assessing the impact of structural modifications in the construction of surveillance network for Peste des petits ruminants disease in Nigeria: The role of backbone and sentinel nodes

**Asma Mesdour**[1,2]*, **Sandra Ijoma**[3], **Muhammad-Bashir Bolajoko**[3], **Elena Arsevska**[1,2], **Mamadou Ciss**[4☯], **Eric Cardinale**[1,2☯], **Mathieu Andraud**[5☯], **Andrea Apolloni**[1,2☯]

1 CIRAD, UMR ASTRE, INRAE, Montpellier, France, 2 CIRAD, UMR ASTRE, Université de Montpellier, Montpellier, France, 3 National Veterinary Research Institute, Vom, Nigeria, 4 ISRA, LNERV, Dakar-Hann, Senegal, 5 ANSES, Ploufragan-Plouzané-Niort Laboratory, EPISABE Unit, Ploufragan, France

☯ These authors contributed equally to this work.
* asma.mesdour@cirad.fr

**Data Availability Statement:** Data are available in zenodo: https://doi.org/10.5281/zenodo.13121488.

## Abstract

Peste des petits ruminants (PPR) is a highly contagious disease affecting mainly sheep and goats. Livestock movements contribute to the spread of the disease by introducing it to naive areas or exposing susceptible animals to it in infected regions. Because of its socio-economic impact, the Food and Agriculture Organisation (FAO) and the World Organization for Animal Health (WOAH) have set the goal to eradicate it by 2030, one of the key steps being the improvement of surveillance networks. The present study aimed to provide tools to identify areas that could serve as sentinel nodes, i.e. areas that may be rapidly infected at the onset of epidemics. Using data from a market survey conducted in Northern Nigeria, we reconstructed the small ruminants mobility network and simulated the diffusion of PPR virus through animal movement. From the analysis of simulation outcomes, we investigated which nodes could act as sentinel nodes under specific conditions for disease transmission. We considered several modified networks to get around the problem of data only being available for part of the overall network structure and to account for potential errors made during the field study. For each configuration, we simulated the spread of PPR using a stochastic Susceptible-Infectious (SI) model based on animal movements to assess the epidemics' extent and the presence of recurrent patterns to identify potential sentinel nodes. We extracted the backbone of the reference network and checked for the presence of sentinel nodes within it. We investigated how the origin (*seed*) of the epidemics could affect the propagation pattern by comparing and grouping seeds based on their respective transmission paths. Results showed that the isolated backbone contains 45% of sentinel nodes that remain stable or undergo only minor changes in 9 out of 11 configurations. On top of that, the characteristics of sentinel nodes identified in the backbone are not influenced by the severity of the disease. The H index, in-degree, and eigenvector are the most essential variables. This study provides an overview of the major axes of animal movements in Nigeria

**Funding:** European Union for the project "EU support to Livestock Disease Surveillance Knowledge Integration" (LIDISKI).

**Competing interests:** The authors have declared that no competing interests exist.

and the most vulnerable locations that should be prioritized for monitoring livestock diseases like PPR.

## Introduction

In sub-Saharan Africa, livestock mobility is essential to the production and trade of livestock and, in turn, is one of the primary sources of income for livestock owners. Due to a lack of infrastructure and storage facilities (like slaughterhouses, fridge cells and warehouses, high-speed roads and adapted trucks), live animals are sold at the nearest local market and then moved through the commercial chain toward larger metropolitan areas for consumption [1]. At the same time, herds are moved around in search of better grazing as part of seasonal and international transhumance practices [1]. In most cases, these movements are international and concentrated between two areas: the arid gridlocked areas, where most of the livestock is concentrated and the humid and greener areas of the West Africa region, where most of the population lives. Transhumance and commercial livestock movements for trade provide red meat for the population while facilitating the spread of animal and zoonotic diseases [2].

Among these animal threats, the Food and Agriculture Organisation (FAO) and the World Organization for Animal Health (WOAH) have targeted Peste des petits ruminants (PPR) for eradication by 2030. In epidemic situations, PPR-related morbidity can reach 90–100%, with a mortality rate ranging from 50–90% (WOAH). In endemic situations, the mortality rate can be around 33–37%. These high morbidity and mortality rates hinder the economic development of affected countries. The PPR Global Eradication and Control Strategy (GCES) comprises four main stages to assess the country's status and activities to be implemented to achieve the status of a PPR-free country [3]. Vaccination is the main control method; however, strengthening the surveillance system is crucial to eradication. In particular, as an output of stages 3 and 4, surveillance systems should be adapted for early warning (stage 3) and focused on the population at risk (stage 4). In recent decades, tools and methods from complex network theory have been massively used in animal and human epidemiology [4]. In its basic representation, livestock mobility can be illustrated as a graph with nodes representing premises and links (often weighted) representing movements between them [1]. Structural analysis can provide preliminary information about potential epidemics' size and propagation speed and identify areas that could facilitate disease propagation, particularly at risk of infection [5]. Further, when stochastic network transmission models are used, information can be retrieved for early warning systems and to determine the spatial extent of the epidemic [6]. Epidemic patterns depend on the interplay between the structure of the networks and the characteristics of the epidemic [7, 8]. The models' assumptions, structure, complexity, and reliability depend on the data quality and degree of detail used to reconstruct the network and the information concerning the disease under study [5]. Even though networks are always characterized by a high degree of heterogeneity and redundancy, there is still a need to understand how the structure of networks influences their dynamics [8]. Identifying likely transmission paths and extracting subgraphs of the network where most information spreads are essential for studying epidemic-spreading phenomena and designing intervention strategies [9]. Indeed, the network backbone is the core component obtained by filtering all the redundant information while maintaining the network hierarchy [9–12]. Nodes of the backbone are potential candidates for surveillance systems. In Nigeria, PPR is enzootic, with seasonal outbreaks mainly occurring during the rainy season (from June to September in the northern Region) [13]. Seroprevalence

varies geographically from a minimum of 11% among animals in the north-western area to a maximum of 41% in the south-western area [14]. Nigeria is one of the main consumers of red meat in West Africa and has to import animals from abroad to satisfy domestic demand. Most likely because of this concentration of transboundary movements, Lineage IV of the PPR virus has been introduced and is now circulating in Nigeria with historical lineage II [14]. However, despite the positive contribution of mobility to animal production, and consequently to Nigeria's GDP [15], and the recognized negative impact of PPR, there is still no system for collecting information and mapping livestock movements in Nigeria. Information on livestock mobility can consequently only be collected through *ad-hoc* activities, such as market surveys. These activities are time- and resource-consuming, restricted to specific areas and a specific period of the year, and only a relatively small number of respondents are surveyed, explaining why only a portion of animal movement can be recorded. Moreover, reticence towards surveyors and/or confusion between official and local names of locations can result in mistaken origins and/or destinations of livestock movements being recorded. Consequently, the real structure of the network may differ slightly, as well as the role that markets play in the diffusion of diseases. In particular, such slight differences may affect the identification of sentinel nodes that could be used as early warning areas for the circulation of the disease. In this work, we extended the analysis conducted in a previous work [16, 17] to identify sentinel nodes. We used data from market surveys in three Nigerian States (Plateau, Bauchi, and Kano) to study the network structure and determine the network backbone. We checked under which network conditions and transmission probability backbone nodes could be used as sentinel nodes. To this end, we developed a stochastic transmission model on the network and ran several scenarios by varying the transmission probabilities or modifying the network structure. We analyzed the different epidemic patterns to identify recurrent ones. We identified sentinel nodes and retrieved their characteristics. In addition, we assessed how sentinel nodes and characteristics change along with variations in the network and increased probability of transmission. We extended the range of transmission probabilities compared with the previous study [16, 17]. We also included an extra scenario of network modifications to understand better the interplay between structural characteristics and transmission probabilities and how these would affect the identification of sentinel nodes.

## Materials and methods

### Data and epidemic simulation

Nigeria is a federal country with three administrative divisions: State, Local Government Area (LGA), and District/Ward. In this study, we used mobility data concerning small domestic ruminants (goats and sheep) collected in surveys conducted in 10 markets in three States in the central and northeastern part of Nigeria Fig 1): 6 in Plateau State (Sabo (Wase District), Tutum (Dengi District), Jarmai (Kantana District), Yelwa (Shendam District), Doemak (Doemak District), Rikkos (Jos Jarwa District)), 2 in Bauchi State (Alkaleri (Pali District), Gadananmaiwa (Ningi District), and 2 in Kano State (Sabongari (Wudil District), Getso (Getso District)). The choice of the markets was based on the outcome of a workshop with stakeholders [18].

In each market, around 100 livestock owners/ traders were interviewed about the origin/ destination of the movements (State, LGA, District, and name of the village), the number of animals of each species, and the reason for the movement. Each market was visited by Dr. Ijioma and local enumerators once between April and September 2022 (one-off). In parallel to the market surveys, focus group discussions were held with actors to complement information. However, this last data set was not included in our analysis, since it is the object of a separate

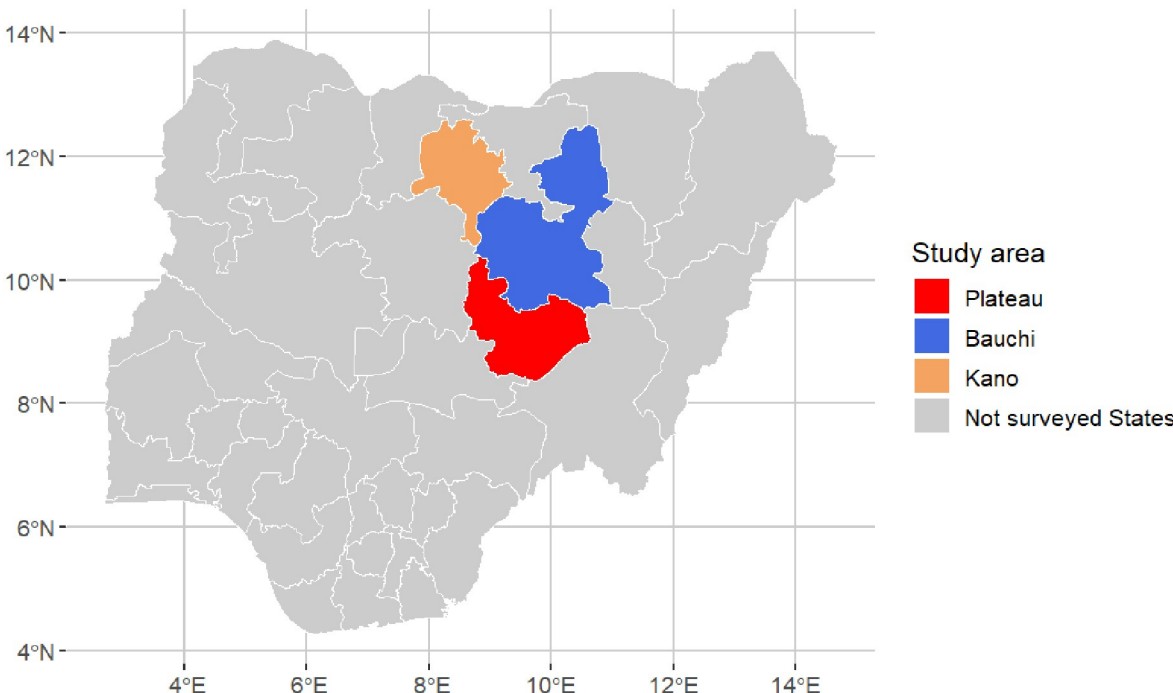

**Fig 1. Map of Nigeria showing our study area.** The colored area indicates the study area, and the gray area shows all States not included in the data collection. Each dot corresponds to a market surveyed: 6 in Plateau, 2 in Bauchi, and 2 in Kano.

work [18]. Locations recorded during the survey were geo-referenced. However, the identification of the villages was difficult because of different factors, among them, the absence of an official database of villages in the area to refer to also for the transcription, the difference between official village names and commonly-used ones, the use of referring to the district to indicate locations. Because of these issues, data were aggregated at the district level (network nodes). A link between two districts was considered if at least one animal was moved between the two districts. The network reconstructed from the data is hereafter referred to as a "reference network" and is used as a reference (configuration A) in the following steps.

The focus of this work is to study the propagation of PPR at the national level to identify possible locations where outbreaks could be reported and assess how the structure of livestock exchanges could impact the diffusion. To this end, we didn't include livestock infection dynamics. In Nigeria, PPR is endemic, with seasonal outbreaks. PPR prevalence varies widely in the countries from 40% to 11%, and the vaccination coverage rate is still low. We considered the inter-epidemic period, during which PPR was re-introduced in the area due to livestock movement. In the following, the term "Infected" nodes indicate nodes where new cases are registered due to the arrival of infected animals from other areas and where transmission occurs to animals in the area. The main parameter $\rho$ encompasses all the dynamics that could lead to the occurrence of at least a case in the area. In this study, we have omitted several disease-related parameters that could impact the attack rate in the local population. This approach has been used to identify candidate nodes for surveillance network [4].

At the beginning of the epidemic, all the nodes are susceptible (S) except one (hereafter denoted the seed) in the infected state (I), chosen among all nodes with a non-zero out-degree. The probability that an infected district could infect a susceptible one through animal movements is denoted $\rho$, and the number of the infected neighbors of node I is denoted $I_i$ [19]; the

probability of a susceptible node I becoming infected (I) follows a binomial distribution, with the probability $\Pi$ defined as follows:

$$\Pi = 1 - (1 - \rho)^{I_i}, \tag{1}$$

In our case, each time step corresponds to one week, the recording frequency used in all the Districts in the dataset. Probability $\rho$ is an "effective probability" that accounts for the different dynamics, for example, that at least one infected animal is being moved between the two nodes, that a contact has occurred between animals and resulted in at least one new infection in the destination node.

Following the work of Herrera [4], we define sentinel nodes as nodes that are infected frequently and promptly during epidemics, i.e., nodes that are most often infected before the peak of incidence is reached. Three factors could affect the propagation of the pathogen and the characteristics of the sentinel nodes: transmission probability, the network structure, and the origin of the epidemic (the seed). To evaluate the impact of transmission probabilities on the epidemic process, we considered nine values of $\rho$ varying from 0.01 to 0.75 (0.01, 0.05, 0.1, 0.15, 0.20, 0.25, 0.3, 0.5, 0.75). For each value of $\rho$, we examined various network configurations (modified network described in the following section). We initiated the epidemic by selecting all seeds with a non-zero out-degree. One hundred simulations of epidemics were run on the observed graph. Then, for each modified network, simulations were stopped when no new infected node was recorded over ten consecutive steps.

Various types of errors can generally be encountered during surveys. In our case, measurement errors were possible because incorrect responses can be given (e.g., village names may have the same name or be mispronounced, leading the interviewer to record a wrong name). Another potential error was sampling error; the sample of interviewees (100 respondents) might not represent all traders passing through the markets where the survey is conducted. We assume the reference network (configuration A) merely represents only a subset of the "actual" animal mobility network, and several movements could not be recorded during the activity (non-observed links) or recorded inappropriately. Our analysis inferred the role of non-observed links and network modifications in disease propagation and the identification of sentinel nodes.

We ran several scenarios with modified networks to account for missing information on the network structures and fill the knowledge gap. This helped assess the impact of modifications in network structure on the spread of the epidemic compared with the observed epidemic results and helped identify movements that impact disease propagation. However, exploring all scenarios could demand a substantial effort, and in this work, we focused on scenarios compatible with existing data collection constraints and in-field situations. As in the previous study [16], we considered two categories of network modifications to assess the impact of typical errors/missing information in data collection:

**Modifications due to structural misinformation.** For several reasons, respondents could be referring to misleading origins/destinations of movements. Among these reasons, we could cite the difference between official names and commonly used names for identifying the location, the absence of an official database of villages, and the reticence of the respondent. This fact, while not affecting the number of movements toward/away from a market, could change the direction of the links. Moreover, only some of the traders in the markets were interviewed, meaning the real number of connections of the nodes may have been underestimated. We thus mimicked changes in trade relationships using random permutations of links or by completing observed data as follows:

1. Random partial reorganization: We randomly reordered 5% (configuration B1), 20% (configuration B2), and 40% (configuration B3) of network links while maintaining the same node indegree for destination locations. This situation corresponds to the case when inaccurate information about the origin of the movements is provided

2. Random link insertion: We randomly added 5% (configuration C1), 20% (configuration C2) and 40% (configuration C3) of network links

**Modifications due to survey limitations.**   Not all markets could be sampled due to limited resources, thus reducing network information about nodes and links. To assess the impact of incomplete data, we simulated networks with reduced sizes by pruning out nodes and links according to two strategies:

1. Removal of central markets: We removed the market with the most connections located in Plateau State (configuration D1) and then removed the market with the fewest connections in Plateau (configuration D2)

2. Removal of peripheral markets: We removed one of the surveyed markets located in Kano (configuration E1) and another surveyed market situated in Bauchi State (configuration E2)

We estimated the average final size of epidemics to assess the extent of simulated epidemics across each configuration and transmission probability. Further, we used Kaplan-Meier models to examine the time to peak infection across various configurations. The models were fitted for each transmission probability ranging from 0.01 to 0.75. Log-rank tests were used to assess differences between configurations.

## Detection of seed clusters

Following a procedure similar to the one used in Bajardi [5], we aimed to identify subsets of seeds (seed clusters) in which epidemics were identical regarding the size and identity of infected nodes. For each network configuration and each seed, we reconstructed the "probable path of transmission" through the following steps Fig 2:

1. given a node $j$ infected at time $t$, we identified all its potential infectors, i.e., neighbors of $j$ node that were previously infected

2. an oriented link was drawn from each potential infector to infected node $j$

3. steps 1 and 2 were repeated for all simulations using different transmission probabilities. The weight of each link in the transmission path is associated with the frequency at which a link appears in all the simulations

The result for each seed was an oriented weighted network called a probable path. Probable transmission paths for different seeds were compared using the weighted Jaccard index [20] to measure the similarity between them and used as a distance measure. The optimal number of clusters was first identified using the elbow method, followed by k-means classification. The clusters detected in the reference configuration (configuration A) and the modified configurations were then compared using the Rand index [21] to assess how modifications could alter propagation patterns. In a second step, the number of nodes reached by seeds in each cluster was compared using analysis of variance, followed by a post-hoc Tukey test, to identify

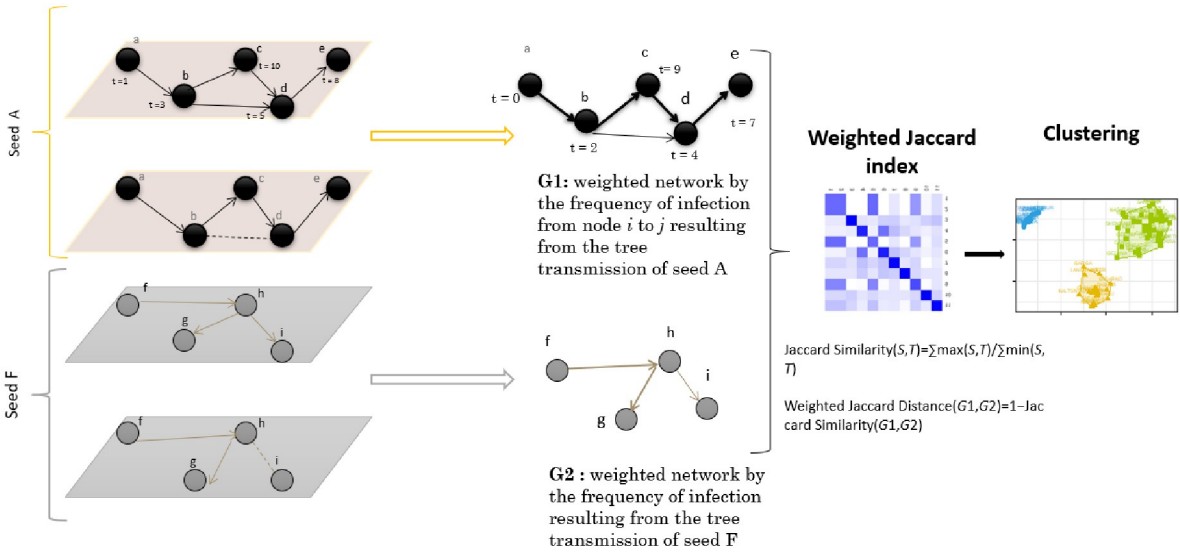

**Fig 2. Steps used to build the probable path of transmission.**

statistically distinct groups. This method allowed the identification of seed clusters, which, once infected, could give birth to large or small epidemics (propagator).

## Node vulnerability and definition of sentinel nodes

In this work, we defined sentinel nodes based on their vulnerability and the moment in time at which they are infected. There is no univocal definition of a node's vulnerability. This article defines vulnerability as the probability of a node getting infected early in the epidemic. We used simulation results to identify sentinel nodes for each configuration and transmission probability. For each node, we estimated:

1. the number of simulations infection occurred before the epidemic peaked *frequency*

2. the average frequency of infection and the standard deviation

3. the classical threshold (average frequency + 2 standard deviations) was used to classify nodes; Nodes with a frequency higher than the threshold were classified as sentinel; otherwise, they were classified as infected (i.e., after the epidemic peaked) and not infected (but remained susceptible throughout the simulation)

## Characterization of sentinel nodes

The above procedure enabled the dynamic identification of sentinel nodes (i.e. through the use of numerical simulations). However, we also wanted to identify the structural characteristics of sentinel nodes. We hypothesized that sentinel nodes can be found among the network's backbone nodes. The backbone is a sparse subset of nodes and links encompassing most information circulating within the network. For this reason, we hypothesize that the disease is most likely spread via the backbone. Several methods have been used to extract the backbone from the network. In this study, we used the LD (Local Degree) approach applied by Neal et al. [12], which assesses edge importance by considering the endpoints of the nodes and prioritizing those linked to central nodes. Following edge ranking and normalization, the most significant

edges of each node are retained in a sparser network that, nevertheless, preserves hierarchical structures. This procedure forms the network by preserving key edges to maintain its essential structure. The backbone was extracted from each configuration. The ten generated backbone structures were compared to the reference backbone (extracted from the reference network) using the Jaccard index to determine its stability.

In this work, we used the following centrality measures to characterize each node in each configuration: in/out Degree [22], betweenness [23], in/out Closeness [23], in/out H index [22], in/out Neighborhood [24], and Eigenvector centrality [25]. We then used a random forest algorithm [26] to classify the centrality measures that characterize sentinel nodes. In addition, we investigated whether the characteristics of the sentinel nodes changed—or not—depending on the probability of transmission by repeating this step for each configuration. A 5-fold cross-validation was done to assess the performance of the random forest model. This entails dividing the dataset into five equal portions. The model is trained on four folds in each iteration and evaluated on the fifth. The process is repeated five times, guaranteeing that each data point contributes to training and evaluation exactly once. By averaging the performances across the five iterations, we obtained a more reliable estimate of the model's predictive ability, thereby reducing the risk of overfitting or bias resulting from a single data split. All analyses were performed using R software (version 4.3.3) and the following packages: ggplot2, igraph, and randomForest.

## Results

### Description of the network and identification of the backbone

The network reconstructed using market data yielded an oriented network with 144 nodes and 268 links, forming one weakly connected component and two strong ones Fig 3. Hereafter, this network is used as a reference for comparisons (configuration A). The backbone extracted from the reference network (reference backbone) comprises 20 nodes Fig 3. Out of the total of 20 nodes, 40% (8 out of 20) are located in Plateau, 25% (5 out of 20 in Bauchi, 10% (2 out of 20 in Kano, and the remaining 25% (5) are distributed across other States, particularly those in the southern part of Nigeria. It is noteworthy that the nodes forming the backbone are geographically dispersed from the northern to the southern States.

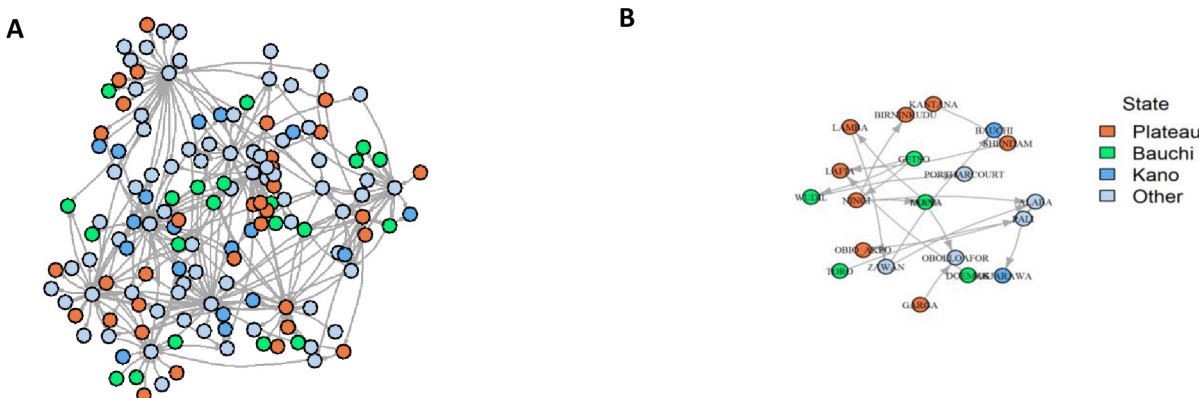

**Fig 3. Visual representation of the reference network (configuration A) and the extracted backbone (reference backbone).** A: Reference network: 144 nodes and 268 links. B: Reference backbone using the sparsify method [11], which contains 20 nodes from different States. Each color corresponds to the State or the District (node) in our study area (Plateau, Bauchi and Kano).

**Table 1. Impact of changes in the configuration on the reference backbone (backbone extracted from the reference network).**

| Configuration | Reduced nodes % | Reduced edges % | Similarity (Jaccard score) |
|---|---|---|---|
| A | 87 | 84 | - |
| B1 | 87 | 84 | 1 |
| B2 | 87 | 84 | 1 |
| B3 | 87 | 84 | 1 |
| C1 | 88 | 85 | 1 |
| C2 | 87 | 88 | 0.2 |
| C3 | 82 | 86 | 0.1 |
| D1 | 89 | 83 | 0.3 |
| D2 | 86 | 81 | 0.9 |
| E1 | 89 | 85 | 0.6 |
| E2 | 90 | 86 | 0.7 |

"Reduced Nodes %" and "Reduced Edges %" are the percentage of nodes and edges that are not found in the backbone for each configuration. "Similarity (Jaccard Score)" is the Jaccard similarity score for each configuration compared to the reference backbone.

The backbone of modified networks was compared with the reference network using the Jaccard Index as a similarity measure [27]; when subjected to modifications, the structure of the backbones for configurations B1, B2, B3, C1, and D2 was identical to the reference one (similarity of 1 or close to 1, indicating an exact match in node composition Table 1). In contrast, the backbone extracted from E1 and E2 has less similarity with the reference backbone (respective scores of 0.6 and 0.7), indicating a moderate difference in the node composition compared to that of the reference backbone. Finally, the backbones of configurations C2, C3, and D1 resemble each other the least, with respective scores of 0.2, 0.1, and 0.3, highlighting divergences from the reference backbone.

## The impact of variations in network structure and transmission probability $\rho$ on epidemic final size

For the reference scenario (configuration A), we simulated 100 simulations for each of the 63 possible seeds of the epidemics. As the in-degree and out-degree remain unchanged, the same number of seeds is found in configurations B1, B2, and B3. Otherwise, the number rises to 70, 88, and 102 in configurations C1, C2, and C3, respectively, and falls to 60, 62, 57, and 58 in configurations D1, D2, E1, and E2 respectively. From 29% to 43% of the seeds are concentrated in Plateau State, while Bauchi State accounts for 20% to 32% of the seeds and Kano State for smaller proportions ranging between 5% and 12%, see S1 Table.

Fig 4 gives an overview of the simulation results obtained for all configurations, focusing on the distribution of final sizes. We used an interval of 20 steps for all the simulations, corresponding to the duration of the rainy season in weeks. Independently of the configuration chosen, the epidemic's final size and duration are influenced by transmission probability, an increase in a shorter amount of time. In all the configurations, the same behavior was observed for the lowest value of the transmission probability ($\rho$=0.01): very few, if not none, of the nodes became infected during the interval simulated. However, when the probability of transmission increased, some structural variations affected the size of the epidemic. The reattribution of links while maintaining the same degree (B1, B2, and B3) or incorporating only 5% supplementary links (C1) did not change the final size with respect to the reference one. We observed a marginal impact when introducing an additional 20% of links (C2) or excluding

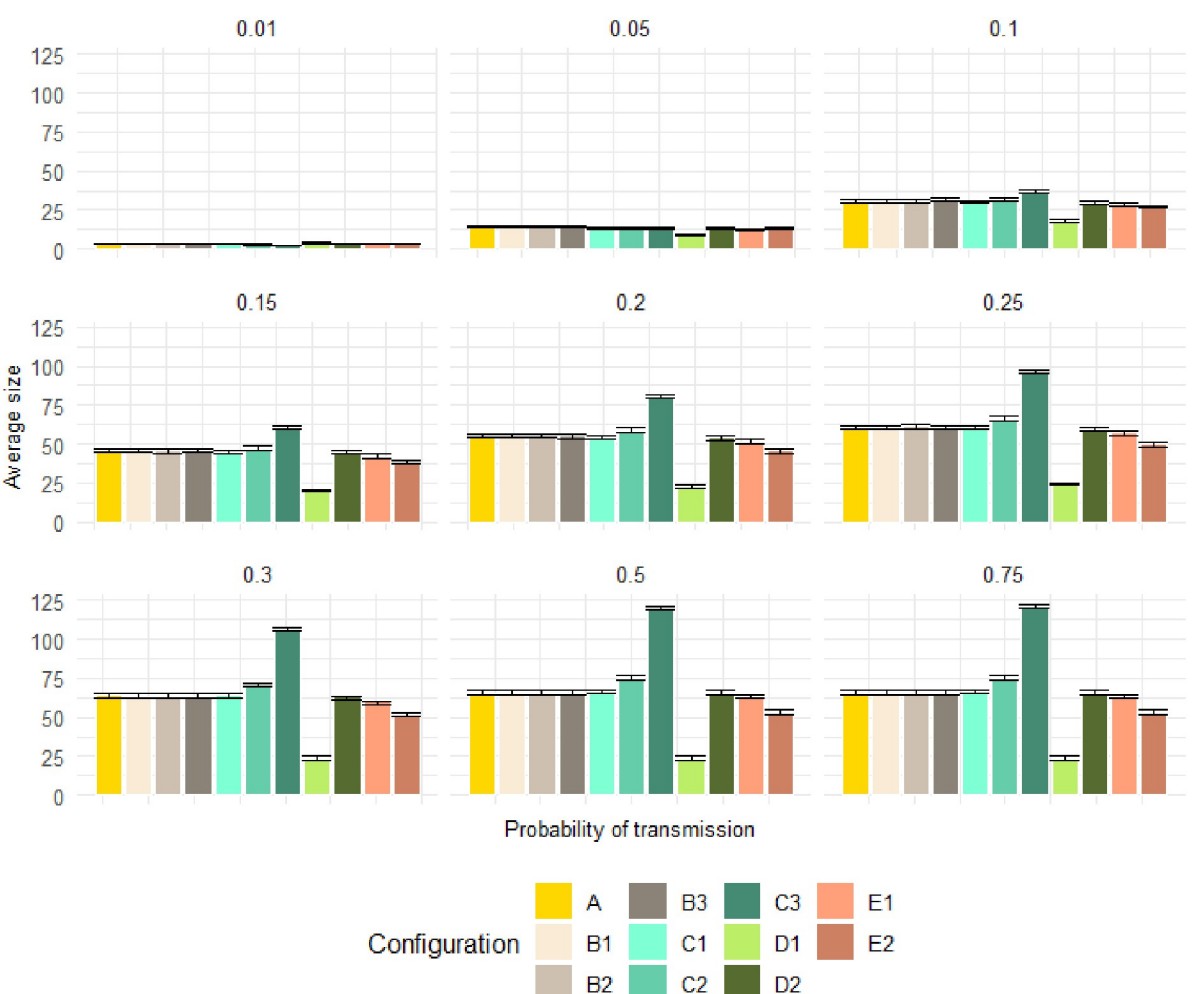

**Fig 4. Results of PPR simulations using the SI model, showcasing the animal mobility network.** The average final size in simulated epidemics is presented under different configurations (from A to E2), and for each transmission probability ($\rho$ = 0.01, 0.05, 0.1, 0.15, 0.2, 0.25, 0.3, 0.5, 0.75). The final size fluctuates significantly under condition C3 (when adding 40% of links) and under condition D1 (deleting the most centrally connected node).

peripheral markets (E1, E2). Adding a large number of links (C3) led to a considerable increase in the final size of the epidemic, from ($\rho$ = 0.1). When we excluded the central market in Plateau State (D1), widely different estimates of the final size of the epidemics were already observed Fig 4 at low transmission probability ($\rho$ = 0.05)) but became clear at higher transmission probabilities.

Fig 5 shows the time required to reach the epidemic's peak. Each plot corresponds to a specific value of the transmission probability, while each color corresponds to a different configuration. As expected, the time to the peak diminishes with increased transmission probability for each configuration. Moreover, significant variations were observed in the time to reach the peak for the transmission probabilities of 0.3, 0.5, and 0.75, with p-values below 0.0001. Significant differences were also observed for transmission probabilities of 0.01, 0.2, and 0.25, emphasizing variability in the time to peak infection based on the network configuration.

Conversely, no significant differences were observed for transmission probabilities of 0.05, 0.1, and 0.15, suggesting comparable infection dynamics across configurations. We used

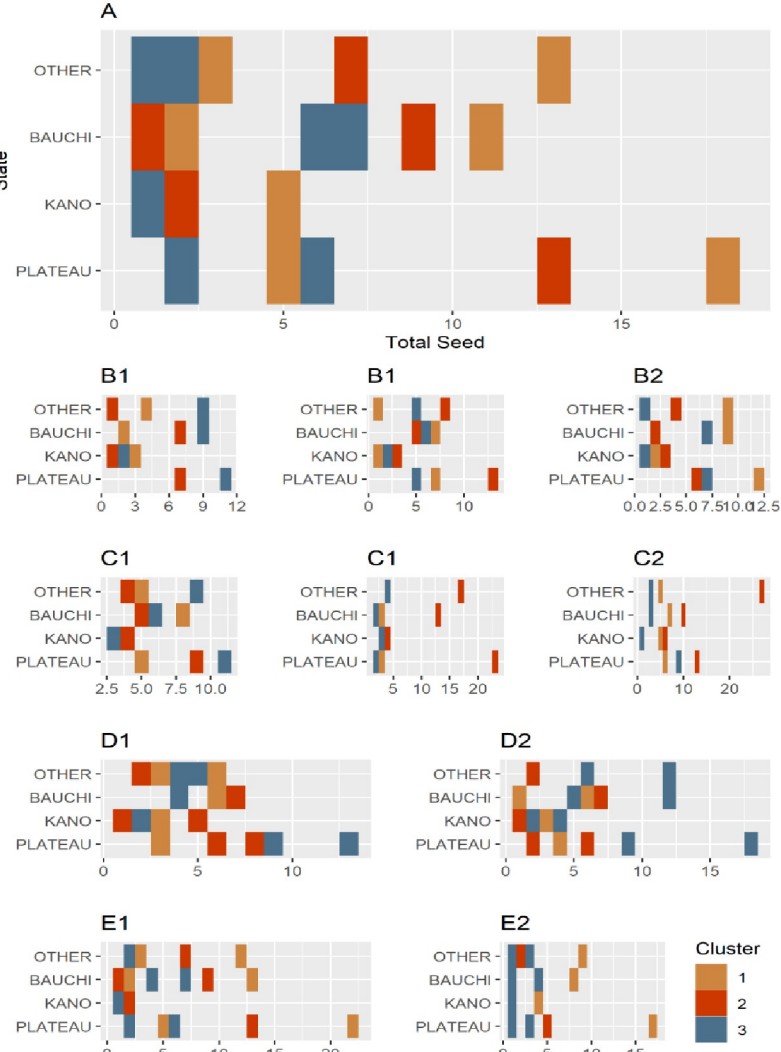

**Fig 5. Kaplan-Meier curves showing the time required to reach peak infection as a function of each transmission probability and configuration.** This model was applied to the SI model's result, including all simulated epidemics. p represents the p-value and $\rho$ the probability of transmission.

cluster analysis to identify a subset of seeds with similar invasion paths and whose epidemics have comparable final sizes. If epidemics originated from nodes belonging to the cluster, they would likely impact the same set of nodes. In the reference network, the elbow method suggested that the optimal number of clusters is three. The clusters consist of geographically dispersed seeds, each formed by seeds originating from different States Fig 6. These observations remained consistent across all configurations.

A comparative analysis of the distributions in the reference network with those in the modified networks revealed that these clusters are sensitive to network modifications. In the B1 configuration, similarity was low (Rand index = 0.45) even in the case of minor variations in link distribution, and this trend persisted with a significant fraction of links rewired in configurations B2 and B3 (0.42 and 0.45, respectively). The similarity in configurations C1 and C2 was comparatively lower (0.56). By introducing more links in the configuration, C3 considerably

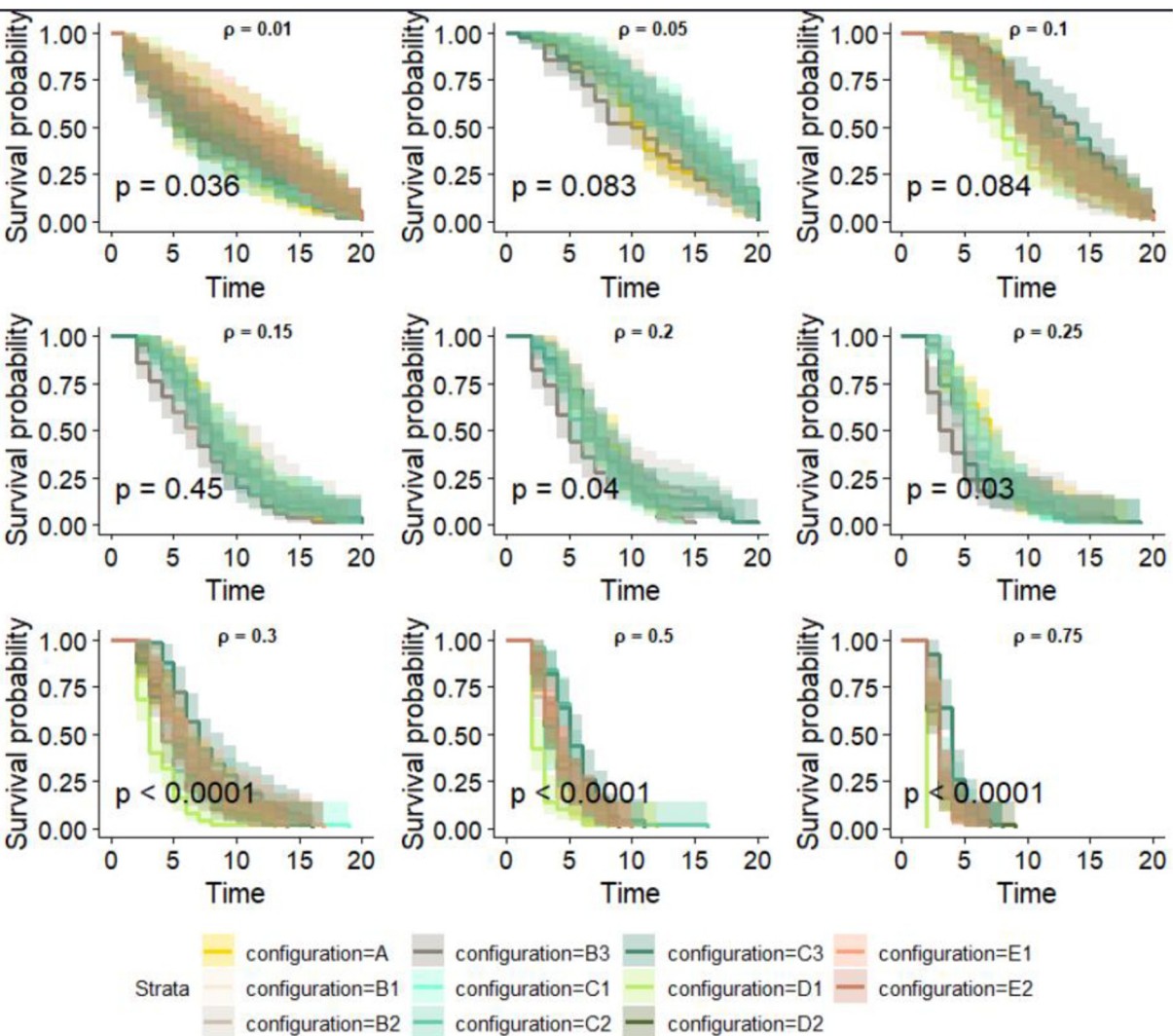

**Fig 6. Number of seeds in each seed cluster per State obtained with the k-means classification.** The number of seeds is shown on the x-axis and the name of the State is on the y-axis. Each line corresponds to the distribution in clusters (color) of seeds in the States.

altered the identified clusters (Rand index = 0.30). Removing a peripheral market in configurations E1 and E2 led to a Rand index of 0.37 and 0.56, indicating changes in clusters particularly identified in configuration E1. However, in configuration D, when the Zawan market, i.e. the most connected in Plateau State, was eliminated, the resulting Rand index was only 0.11. Otherwise, eliminating the least connected central market has less influence (0.56) on cluster distribution than in configuration D1. Analysis of the size of the epidemics revealed a significant influence of the cluster in all configurations (p-values <0.0001), indicating significant statistical differences in epidemic potential among clusters and that, depending on the cluster to which the seed belongs, epidemics could have different extents. Tukey's test revealed significant differences (p < 0.01) in the size of the epidemic among the clusters, indicating that some clusters tend to infect more nodes than others, see S2 Table. In the following, "cluster propagator" refers to the cluster whose epidemics originating in one of its seeds reaches the largest

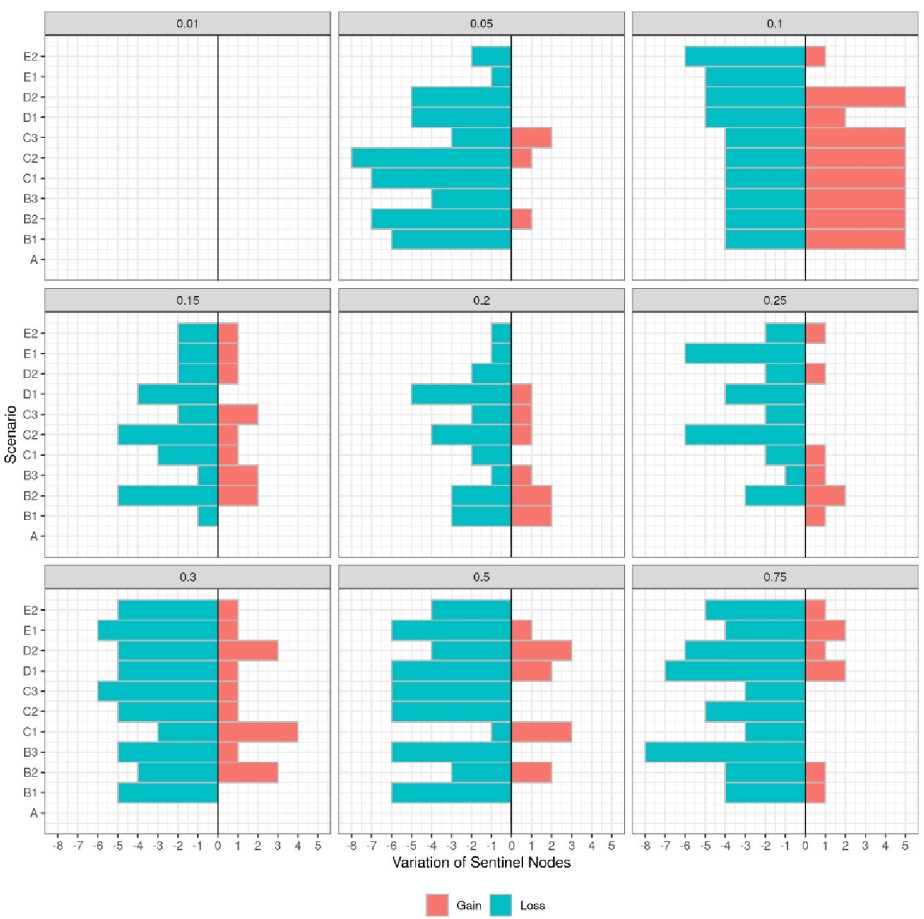

**Fig 7. Tornado plot showing the variation in the number of sentinel nodes for each transmission probability.** The gain or loss of the number of nodes according to the reference (here, it's $\rho$ = 0.1) is shown on the x-axis, and the different scenarios used in the study (configurations) are displayed on the y-axis.

number of nodes. Across all configurations, most seeds in the cluster propagator belonged to Plateau State, followed by Bauchi, while only a small number belonged to Kano or other States.

## The number and identity of sentinel nodes varied with $\rho$ and configuration

Fourteen sentinel nodes (Alaba, Dengi, Gwaywalada, Jos Jarawa, Lamba, Obolloafor, Okoamako, Pali, Port Harcourt, Shendam, Taura, Zawan, Lafia, and Wase) were identified in the reference network for various transmission probabilities ($\rho$ > 0.01). Notably, three nodes—Obolloafor (Enugu State), Okoamako (Delta State), and Port Harcourt (Rivers State)—consistently maintained their vulnerability status independently of the transmissibility of the disease. When we examined the results of the different transmission probabilities separately, at $\rho$ = 0.01, no potential sentinel node was identified, but from $\rho$ = 0.05, between 6 and 9 nodes exhibited vulnerability per transmission probability Fig 7. The total number of sentinel nodes in the other configurations was lower than in the reference network (B1 = 9, B2 = 11, B3 = 10, C1 = 9, C2 = 8, C3 = 11, D1 = 7, D2 = 10, E1 = 9, E2 = 8). However, it is worth noting that five nodes (Alaba, Jos Jarawa, Obolloafor, Okoamako, and Wase) consistently emerged as sentinels and remained fixed across all configurations.

**Table 2. Number of sentinel nodes shared between configurations and the backbone, with State-wise specification.** CommonRef indicates the set of nodes in the reference backbone (i.e. configuration A) that are also sentinel ones. Sentinel nodes are counted independently of all transmission probabilities.

| | Number of common nodes | Plateau State | Bauchi State | Kano State | Other State |
|---|---|---|---|---|---|
| CommonRef | 9 | 5 | 1 | 0 | 3 |
| CommonRef ∩ B1 | 7 | 3 | 0 | 0 | 4 |
| CommonRef ∩ B2 | 8 | 3 | 1 | 0 | 4 |
| CommonRef ∩ B3 | 8 | 4 | 0 | 0 | 4 |
| CommonRef ∩ C1 | 8 | 4 | 1 | 0 | 3 |
| CommonRef ∩ C2 | 7 | 3 | 1 | 0 | 3 |
| CommonRef ∩ C3 | 9 | 4 | 1 | 0 | 4 |
| CommonRef ∩ D1 | 6 | 2 | 1 | 0 | 3 |
| CommonRef ∩ D2 | 9 | 4 | 1 | 0 | 4 |
| CommonRef ∩ E1 | 9 | 4 | 1 | 0 | 4 |
| CommonRef ∩ E2 | 8 | 4 | 1 | 0 | 3 |

### The presence of sentinel nodes on the backbone

As mentioned above, the reference network contained 14 sentinel nodes, nine on the backbone, representing 45% of the backbone nodes. Between 6 and 9 nodes were shared between the reference backbone, the sentinel nodes of the reference network, and the sentinel nodes of each configuration. Table 2) shows that most shared nodes are located in Plateau State, while only one shared node is located in Bauchi State and none in Kano State. The remaining sentinel nodes are distributed across other States (mostly in southern states).

We used random forest classification to analyze the centrality measures that characterize sentinel nodes. Fig 8 presents the Gini index, highlighting the most critical centrality measures across all configurations within each transmission probability. In Configuration A, the in-degree centrality and in-H index appear to be the most critical factors (Gini Index between 60 and 100%). The importance of eigenvalue centrality increased with an increase in transmission probability and became the most crucial factor for $\rho = 0.75$. As defined in [26], the eigenvalue centrality indicates the node's importance, which is expected since structural properties take the lead for $\rho$ close to 1 and consistently maintain a high value across all transmission probabilities, indicating a robust influence of incoming connections. On the other hand, in configurations B1, B2, and B3, in-neighbourhood and in-closeness appear to be essential centrality measures in addition to those observed in the reference network. In configurations C1, C2, and C3, in-neighbourhood, in-degree, eigenvector, and in-closeness are the most important, and their Gini index varies with transmission probability. The in-degree and in-H index are the most important in configurations D1 and D2. Next, in-neighborhood, eigenvector, and in-closeness appear relevant but vary with the different probabilities. Finally, configurations E1 and E2 parallel configuration A in high in-degree centrality but vary in eigenvector centrality, in-H index, and in-closeness. However, when the sentinel nodes on the backbone were restricted, the essential characteristics of the sentinel nodes remained the same even when transmission probabilities varied Fig 8. These attributes include in-degree, in-H index, and eigenvector centrality. Significantly, they retained the same importance and order concerning different transmission probabilities.

## Discussion

In Nigeria, the lack of a livestock identification system and an automatic data centralization system are obstacles to depicting a reliable network of commercial livestock mobility [27].

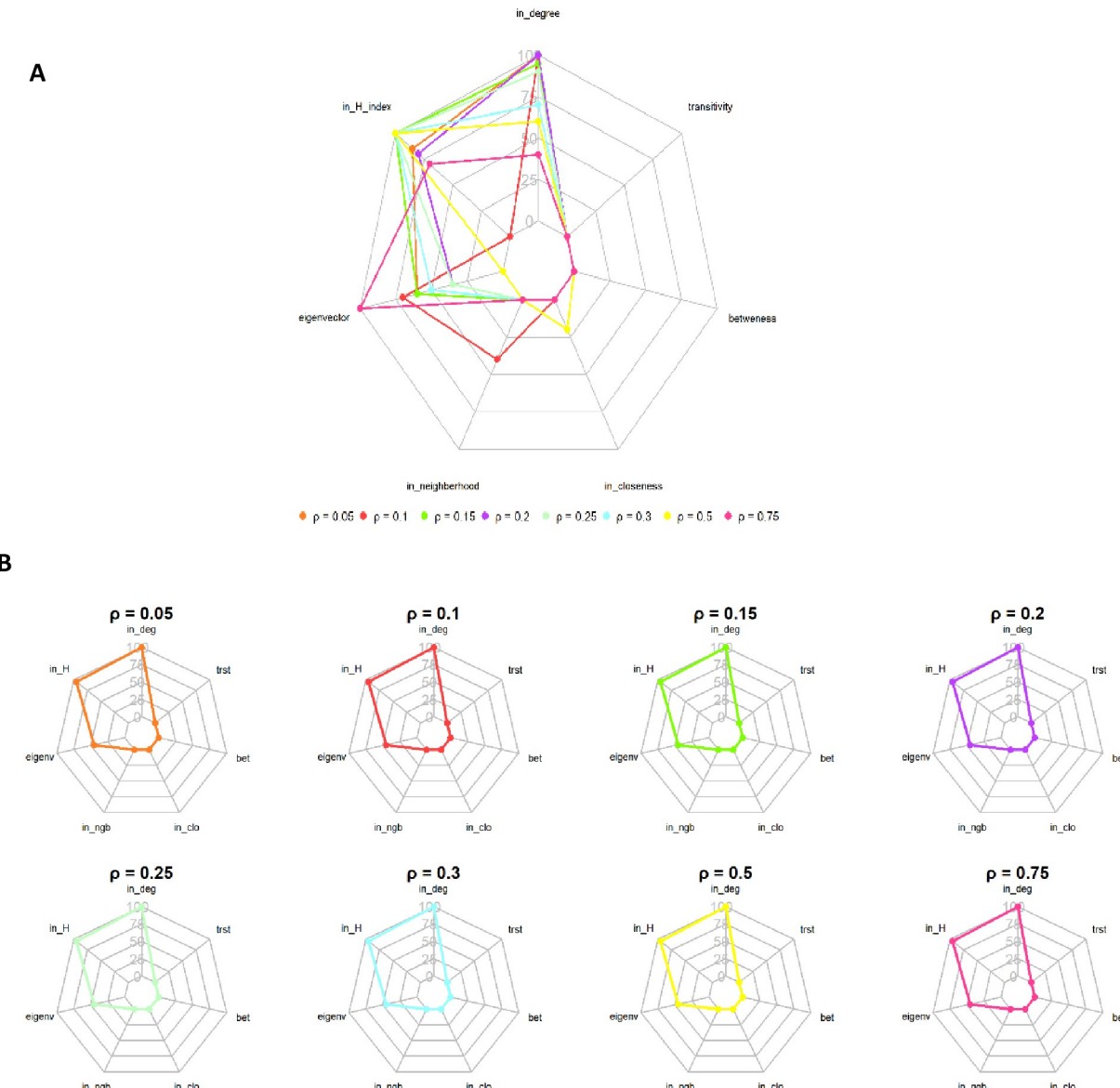

**Fig 8. Random forest result.** A: Gini Index showing the most important centrality measure per probability of transmission of the observed network. B: Most important centrality measure per probability of transmission of the sentinel node on the backbone. The result was obtained from the random forest classification. *Iin_deg* = in-degree, *in_H* = in-H index, *eigenvec* = eigenvector, *in_ngb* = in-neighborhood, *in_clo* = in-closeness, *bet* = betweenness, *trst* = transitivity.

Such data are essential to pinpointing sentinel nodes and, hence, designing surveillance and control programs. This study primarily focused on identifying nodes likely to be used as sentinels across three Nigerian States, using the network we built using data gathered in a market survey as a reference network and a stochastic SI model to simulate the diffusion of PPR. To overcome the reference network's limitations and assess the impact of unobserved links and nodes, we generated ten distinct configurations using elementary link operations such as rewiring and addition and targeted node deletion. The impact of structural modifications on the dissemination of epidemics became apparent when 40% of links were added, and resulted in faster and more extensive diffusion. However, eliminating the peripheral market in Bauchi

State influenced the diffusion dynamics. This finding aligns with the insights offered by Wright [28], whose research highlighted the pivotal role of peripheral nodes in the intricate dynamics of directed networks—eliminating the most highly connected central markets, such as the Zawan market (one of the markets surveyed), can significantly impact the final size of the epidemic. This emphasizes the critical need for market selection during sampling to avoid losing information crucial for effective surveillance and control measures.

For each configuration, we identified clusters of seeds based on the similarity of their invasion path. Clusters were spatially scattered and present in several States. This observation aligns with the results reported in [5]. A cluster reaching more nodes (called seed propagators) was identified in each configuration, whose members are mostly located in Plateau State. This can be explained by the position and economic activity of Plateau State, the gateway for animal movements towards the densely populated area on the coast [15]. Few seed propagators were identified in Kano State. However, this result should be interpreted cautiously, as it could be due to non-homogeneous sampling, as only two markets were sampled in Bauchi and Kano. In contrast, six markets were sampled in Plateau State. Increasing the sample size in States such as Bauchi and Kano is feasible using either a randomized or stratified sampling approach. This will enhance the representativeness of the markets selected for the study.

The study successfully located 14 sentinel nodes in the reference network, dispersed in 9 different States: Alaba (Lagos State), Dengi (Plateau State), Gwaywalada (Federal Capital Territory), Jos Jarawa (Plateau State), Lamba (Plateau State), Lafia (Nasarawa State), Obolloafor (Enugu State), Okoamako (Delta State), Pali (Bauchi State), Shendam (Plateau State), Wase (Plateau State), Zawan (Plateau State), Taura (Jigawa State), Port Harcourt (Rivers State). Six of the 14 nodes are located in Plateau, underscoring the region's susceptibility to diseases like PPR. Regardless of transmission probability, the status of three nodes, Obolloafor, Okoamako, and Port Harcourt, remained invariant. These nodes could thus function as optimal sentinel nodes. Five of the ten markets sampled were classified as sentinel, four located in Plateau State and 1 in Bauchi State. This finding raises questions about the representativeness of the chosen sample and suggests bias in the network reconstruction as the network was built using data collected in these markets. Nevertheless, other sentinel nodes that do not belong to the market were identified, indicating that although the quantity of data collected is limited, some information on network structure can be captured. In other configurations, the number of sentinel nodes decreased slightly. Thus, adding links does not necessarily increase the number of sentinel nodes; the identity of the sentinel nodes varies. Only five nodes consistently kept the same status across all these configurations, of which 2 (Obolloafor and Okoamako) are the stable nodes mentioned previously. These nodes are not among the markets sampled so that they could be the best sentinel nodes.

By isolating the backbone, we gained a clearer understanding of the core structure of the network. The finding that 45% of the sentinel nodes are located on the backbone is a significant insight. It suggests that, as a structurally important subset of the network, the backbone plays a crucial role in harboring sentinel nodes. From a broader perspective, this finding implies that monitoring and protecting the backbone may be particularly important to prevent the risk of early disease spread. More research is needed to understand the exact role of the other half of the backbone nodes that are unsuitable sentinel candidates. In addition, a more appropriate method to better capture vulnerability can also be explored.

The comparison between the backbone extracted from the reference network and the sentinel nodes identified in the observed network across all configurations revealed a consistent alignment with 6 to 9 common nodes. The backbone can be a reliable proxy for identifying sentinel nodes and transcending network fluctuations. However, our study underlines that modifications related to link reattribution failed to alter the structure of the backbone; indeed,

even adding up to 5% of links did not significantly influence its structure. However, beyond this threshold, the structure did undergo drastic changes. This suggests that up to 5% missing data may produce consistent results, but more than 20% missing data will render the backbone structure less stable. If a central market with the least connections is not sampled, it will not impact the backbone structure, while the opposite may be true if the most connected central market is omitted. What is more, the removal of peripheral markets had only a moderate impact on the structure. Thus, using the backbone as a set of monitoring nodes may be adequate when disturbance is limited. Still, beyond a certain number of modifications, increased caution should be exercised in its use for monitoring purposes. In-depth sensitivity analysis is required to accurately identify the exact thresholds and types of disturbance that significantly impact the reliability of the backbone. Presently, the most influential centrality measures in defining sentinel nodes differ across configurations and reveal sensitivity to changes in transmission probabilities. Overall, the in-degree and in-H index appear to be the best indicators of sentinel nodes in the reference network. At the same time, the roles of the eigenvector, in-closeness, and in-neighbourhood were more nuanced for other configurations. They depended on the interplay between the structure of the network and transmission probabilities. In-degree played a pivotal role in constructing our SI model, thus justifying its relevance as a significant feature. The in-H index [22] has been shown to play a pivotal role in identifying sentinel nodes, therefore serving as a crucial metric to characterize sentinel nodes within a network. Derived from the broader concept of the H-index and commonly used to assess a researcher's productivity and academic impact [29], the in-H index focuses on node relationships in the context of networks. A node with a high in-H index implies that it is surrounded by neighbors that are potential sources of infection. In other words, a high in-H index suggests that the node is at the core of a network with diverse neighbors, thereby increasing the likelihood of rapid infection. This metric provides vital insights into the network structure, helping identify nodes that may be particularly influential in the potential diffusion of disease.

In our study, the in-degree and in-H index are correlated in all the configurations, highlighting the resilience of these centrality measures. While not a major concern for the random forest model, which adeptly manages correlated variables, the potential redundancy of information between these measures demands more attention since it may introduce complexity in attributing importance to each variable in the model. Targeting nodes with high in-degree and in-H index can make PPR surveillance cost-effective by concentrating resources on these critical areas that are more likely to detect disease timely. The outcomes of this work would help focusing surveillance efforts where they are most needed, optimizing the use of resources and reducing the costs associated with widespread monitoring. To implement it and made it sustainable, several strategies can be employed: redistributing veterinary officers and/or Community Animal Health Workers (CAHWs) in these sentinel areas to report routinely information about the epidemiological situation; involving communities through engagement and training programs to enhance local participation and reporting; developing automated data collection platforms fed by the veterinarian officers that could provide alerts on realtime to streamline operations.

Eigenvector has been highlighted as one of the essential measures, but it needs to be more stable across configurations and transmission probabilities. Analyses conducted by Herrera [4] and Colman [30] already demonstrated the ability of eigenvector centrality to identify sentinel nodes. Nevertheless, its variable influence may depend more on specific network conditions and transmission probabilities; therefore, caution should be exercised when selecting it as an attribute for a sentinel node.

In-closeness and in-neighborhood appear to be important in several cases, indicating that the influence of overall network proximity and direct incoming neighbors can influence

vulnerability. These measures can be considered indicators of vulnerability. Still, the variability of their importance per probability of transmission and configuration requires a thorough evaluation of their relevance for identifying sentinel nodes.

However, our analysis of backbone characteristics showed that the attributes of nodes identified within the backbone remained constant across different transmission probabilities. In-degree, in-H index, and eigenvector centrality were the most significant. This suggests that extracting a backbone and estimating these measures could provide a set of nodes to monitor closely.

While implementing prevention and control strategies targeting nodes with a high in-degree and/or in-H index may prove effective, the complexity of calculating these measures—mainly due to the scarcity of data and temporal considerations—means that we should not rely solely on these identified criteria. Optimal choices for sentinel nodes depend on several factors, including the network's structural layout (e.g., nodal positioning) [30], the flow dynamics within the network, disease transmissibility (with less transmissible diseases posing tracking challenges), and temporal variations within the network (nodes with stable contact sets are better suited for epidemic detection) [31]. However, given that network structure can evolve due to seasonal transhumance, the movement of livestock linked to festivities, and the presence of diseases in the area, methods are required to adjust the criteria used for sentinel node identification in response to changes. This can be achieved by using dynamic approaches to be sure that the identification of sentinel nodes remains relevant and reliable despite network fluctuations.

In this work, we considered only three cases of network modifications, i.e. when a low (5%), medium (20%), and high (40%) proportion of links were added/rewired. A more systematic analysis, considering more modification levels, will be necessary to determine the threshold at which structural changes become critical and impact disease progression. This study has underlined that adding links impacts the backbone structure when more than 20% of links are added. Yet, the specifics of these changes and the nuances resulting from adding between 5% and 20% links still need to be explored. As the quality of the prediction depends on the epidemic parameters, such as transmission and recovery rates, a more sophisticated model (SIR) would be capable of better capturing the interplay of animal movements (structure, volume, and temporal aspects) and transmission dynamics. To this end, more data should be collected to ascertain whether these same structural characteristics will continue to justify node vulnerability. Additional analysis is required to validate the model, compare disease simulation outcomes on the data-based and reconstructed networks, and cross-reference these with biological information such as PCR data.

## Supporting information

**S1 Table. Number of nodes (districts) in all configurations taken as seed per States.**
(DOCX)

**S2 Table. Comparison of the average final size of the epidemic between the cluster identified by the k-means methods using Tukey test.** diff: The difference between the means of pairs of cluters. lwr: Lower limit of the confidence interval for the difference. upr: Upper limit of the confidence interval for the difference. p adj: The adjusted p-value for each pair of clusters.
(DOCX)

## Acknowledgments

We want to express our heartfelt gratitude to the field team whose efforts made the data collection for this study possible, the National Veterinary Research Institut (NVRI) of Nigeria for hosting. We also thank all the livestock traders who participated in the study. Their cooperation and willingness to share information were crucial in obtaining valuable insights and completing this research. Finally, we thank the conference complex network and their application for inviting us to publish this work.

## Author Contributions

**Conceptualization:** Andrea Apolloni.

**Data curation:** Sandra Ijoma, Muhammad-Bashir Bolajoko, Andrea Apolloni.

**Formal analysis:** Asma Mesdour.

**Methodology:** Asma Mesdour, Mamadou Ciss, Mathieu Andraud, Andrea Apolloni.

**Supervision:** Eric Cardinale, Mathieu Andraud, Andrea Apolloni.

**Validation:** Asma Mesdour, Mathieu Andraud, Andrea Apolloni.

**Visualization:** Asma Mesdour.

**Writing – original draft:** Asma Mesdour, Elena Arsevska.

**Writing – review & editing:** Mathieu Andraud, Andrea Apolloni.

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
