## [Decision Letter · Decision Letter 0]

11 Jun 2024

PONE-D-24-11256Assessing the impact of structural modifications in the construction of surveillance network for transboundary animal diseases: the role of backbone and sentinel nodesPLOS ONE

Dear Dr. MESDOUR,

Thank you for submitting your manuscript to PLOS ONE. After careful consideration, we feel that it has merit but does not fully meet PLOS ONE’s publication criteria as it currently stands. Therefore, we invite you to submit a revised version of the manuscript that addresses the points raised during the review process.

The reviewers recommend that you make improvements to the manuscript. Please attend to all the comments and return the improved manuscript as advised in this letter.

We look forward to receiving your revised manuscript.

Kind regards,

Martin Chtolongo Simuunza, PhD

Academic Editor

PLOS ONE

Journal Requirements:

"European Union for the project “EU support to Livestock Disease Surveillance Knowledge Integration” (LIDISKI)"

"The research was supported by a grant (FOOD/2019/410-957) from the European Union for the project “EU support to Livestock Disease Surveillance Knowledge Integration” (LIDISKI)."

"European Union for the project “EU support to Livestock Disease Surveillance Knowledge Integration” (LIDISKI)"

7. We noted in your submission details that a portion of your manuscript may have been presented or published elsewhere. "This publication is an extended version of a conference proceedings "Complex network & their application 2023", published in the procedeeing, which invited us to submit an extended version of the same article on Plos One." 

8. In the online submission form, you indicated that "access to data on request to Sandra Ijoma and Andrea Apolloni, article co-author"

9. Please amend the manuscript submission data (via Edit Submission) to include author Sandra Ijoma, Muhammad-Bashir Bolajoko, Elena ArsevskaMamadou Ciss, Eric Cardinale, Mathieu Andraud.

10. Please include captions for your Supporting Information files at the end of your manuscript, and update any in-text citations to match accordingly. Please see our Supporting Information guidelines for more information: http://journals.plos.org/plosone/s/supporting-information.

Reviewers' comments:

Reviewer's Responses to Questions

**Comments to the Author**

1. Is the manuscript technically sound, and do the data support the conclusions?

Reviewer #1: Partly

Reviewer #2: Yes

2. Has the statistical analysis been performed appropriately and rigorously? 

Reviewer #1: No

Reviewer #2: Yes

3. Have the authors made all data underlying the findings in their manuscript fully available?

Reviewer #1: Yes

Reviewer #2: No

4. Is the manuscript presented in an intelligible fashion and written in standard English?

Reviewer #1: Yes

Reviewer #2: Yes

5. Review Comments to the Author

Reviewer #1: The study “Assessing the impact of structural modifications in the construction of surveillance network for transboundary animal diseases: the role of backbone and sentinel nodes” set out to identify critical districts in which to institute surveillance activities among other control activities of TADs with a special focus on Peste des petits ruminants (PPR). The study used data from 10 market surveys in 3 states and employed a mix of standard social network analysis and mathematical modelling approaches to identify the most probable “sentinel districts” in Nigeria. The authors present a strong case in regard to the urgent need to effectively control PPR and eventually eradicated through activities such as building strong and efficient surveillance systems, a key contribution to the global efforts to eradicate PPR.

However, the manuscript in current state suffers a few serious methodological and minor scholarly considerations that the authors should consider attending to so as to achieve their set objective (s).

General comments

The authors conveniently define ‘sentinel nodes’ as those areas that are rapidly infected first in case of disease introduction, a definition that leans more to anatomy (cancer research). I would have thought that since the focus is control of TADs, a definition that is borrowed from epidemiology is more appropriate - sentinel nodes as to mimic sentinel monitoring or surveillance. These would be those districts intentionally identified to enable quick disease detection and patterns that guide more conclusive epidemiological investigations to determine spread mechanisms. For example districts that have a very high in-degree or high betweenness would be very good sentinels for disease surveillance because they simply receive animals from various origins that could potentially introduce disease.

Much as the authors apply robust techniques to identify the network backbone that has high potential for surveillance activities a much clearer approach for identifying sentinels for surveillance should have considered. For example, build the network and identify communities (if they exist) and each community selected districts with the highest selected centrality measure (s) – influential nodes and then those become sentinel nodes. Sentinel nodes would make much more sense if the authors then explored them over time (longitudinally) as cross-sectional view of the sentinels doesn’t tell a full picture. The authors need to explain more about their choice of approach to deal with missing/unreliable survey data especially inline with naural network structure of livestock mobility in Nigeria

In the epidemic simulation, the authors chose a “district” as the unit to be susceptible or “infected” and simulated the speed and breadth of PPR penetration across the different nodes. The authors need to elaborate more why they ignored other important epidemiological parameters that affect PPR transmission such as the small ruminant population density in each district, vaccination coverage, PPRV “spreadablity” (R0), animals that recover may not be “infectious” in the next cycle of infection etc. A suitable explanation needs to be included such that this study is more reliable.

Specific comments

The title needs to be revised to include PPR or Nigeria (or both). Or if the title will remain as is, the introduction section of the abstract needs to start off with more general literature about livestock mobility, sentinel nodes etc as PPR seems sneaked in the introductory paragraph of the abstract (given that it is not anywhere in the title).

Lines 6—10: Please crosscheck with the journal guidelines on author affiliation addresses otherwise in their current form they are abbreviated to the extent that one can hardly figure them out

Line 15: please rectify this ‘sweeping’ statement “....and is transmitted through livestock movements” because it not entirely true

Line 20: rework that sentence as it is difficult to comprehend

Line 42: a lot of general information is included here without supporting reference. For example, lack of storage facilities – the authors need to state what that means in an ambiguous manner, do you mean fridges/freezers, warehouses?

Line 47: please consider introducing the abbreviations in full and then only use them on subsequent mentions. For example, what does “WHOA” stand for? Additionally, thrught the entire manuscript, please write Peste des Petits Ruminants as sentence case (only capitalise the first letter), as it is just a name (noun) of a disease, so only the first letter should be capitalised

Line 49: high morbidity and mortality rates: mention some figures and give credit to the sources (as “high rates” does not communicate much)

Line 50: The PPR eradication strategy actually stresses vaccination as the main-stay control intervention. Consider adding a sentence to this paragraph to give non specialist readers more context about the eradication plan

Line 109: This is the first figure, but it labelled figure 8. The figure numbers need to be harmonised throughout the write up. And what state is represented by color grey? Please rework the figure legend

Line 111: please state clearly how many farmers exactly were interviewed per market and was this survey conducted in the same month, repeated observations or just one-off etc? Also consider replaced “were questioned” with “interviewed”

Line 114: “and to avoid problems related to the misspelling of village names” , the authors need to explain why they were unable to recruit local enumerators who could correctly spell the names of villages. Otherwise, this sounds like no one can correctly spell names of villages in Nigeria

Line 119: the authors need to include an explanation for their assumption of district susceptibility without considering vaccination coverage, animal population density, recovery from PPR confers protection (and animals may or may not be infectious to others etc). Why these assumptions were not included in the model needs to be explained

Line 141: more elaboration is needed on “non-observed links”, what are they and how can they be identified without necessarily observing them

Line 147: the sentence is vague. Consider re-writing it in a more concise manner

Line 151: how was ‘misinformation’ judged? What gold standard did the author use to determine that the responses from the survey contained “inaccurate information”. This needs to be made clear in the methodology. Again, state how many farmers (individual or groups of farmers) were interviewed as opposed to stating that “only some of the farmers” were interviewed

Random permutations do not seem to mimic natural small ruminant movement network structures. For example randomly re-ordering links may not necessarily mimic how naturally the “sources of animals eg rural areas” to “urban destination because that is where demand for livestock and their product is” – may not be a scenario one can just assume that the direction of flow can randomly be reversed. In my opinion, in addition to all the innovative approaches employed by the authors to correct for missing/misleading information:

The authors needed to use their market survey data and triangulate it with local key informants with subject specific knowledge to validate the reposes from the surveys. Additionally, the authors should have explored other mechanisms of data collection such as participatory mapping exercises with key stakeholders to reconstruct reliable and robust movement networks that are closer to the true picture in Nigeria (as has previously done by other researchers eg <https: 10.1038="" doi.org="" s41598-023-35968-x="">, <https: 10.1098="" doi.org="" rstb.2018.0264=""> ). Otherwise, the authors need to clearly state the epidemiological assumptions behind their choice of correction of missing data

Line 177: Once again it should be stated whether ‘frequency of infection’ takes care of the fact that infected animals in one epidemic may not be “infected again” as those animals will no longer be susceptible and thus move into the next class of “recovered”. It is not clear if such assumptions were considered in the simulation exercises. Additionally, for districts with not-large-enough populations of small ruminants, the epidemic will naturally die out once susceptible individuals are exhausted, and therefore I would expect some consideration of small ruminant density as a parameter. The authors need to provide an explanation or at least acknowledge that they are aware

Line 231: why did the authors expect the characteristics of nodes (locations eg districts) to change with increasing probability of infection? And also because some nodes were removed , edges re-wired in the different configurations, some changes were really expected even before the analysis. The question is; how do these configurations mimic the natural network structure of small ruminant movement in Nigeria (both static and dynamic)

Line 286: figure legend needs to be rewritten so as to clearly communicate what the fiure represents

Lines 354: The manuscript is silent about “Data analysis” section eg how the analyses were done, what software tools etc. In addition to the random forest plots, consider using the effect size and then explore visualizing the plots with interpretation of what metrics has the largest or smallest effect sizes (a suggestion)</https:></https:>

Reviewer #2: Excellent piece of research. - well described methods and results. Noting i am not a high level expert in network analysis this was helpful!

i would like to see some inclusion of the application in the discussion. Sentinel surveillance is cost-effective and if targeted appropriately contributes very relevant data to overall disease control Could tools be developed to make this sustainable for example?

I was not able to access a key reference - no.15 to ensure complementarity rather than duplication/permissible overlap.

Some minor suggestions/edits:

L31 45% 'of the' sentinel

L40 "pillar" .. Better 'essential' or similar wording

L45 "unify a region" unclear -

L37 WHOA --- WOAH

L51 "massively" - suggest 'extensively'

L60 data's to data

L68 "and circulating" not needed

L73 "of PPR" … `PPR virus'

L74 "line" to 'lineage

L77 'ad hoc' (italics)

L88 "and if so" - doesnt read well - delete or reformat sentence

L147 "tantamount" - paramount / supreme / substantial

L286 Fig 2 caption on different page to figure

L295 Fig 3 caption on different page to figure

L355 Gini

References

Ref 2 "bavf"?

Duplicate refs 30/31

Replace in english for "disponible sur.." and names of months

6. PLOS authors have the option to publish the peer review history of their article (what does this mean?). If published, this will include your full peer review and any attached files.

Reviewer #1: **Yes: **Joseph Nkamwesiga

Reviewer #2: No

---

## [Author Response · Author response to Decision Letter 0]

29 Jul 2024

4. We note that the grant information you provide in the 'funding' and 'financial disclosure' sections do not match

The research was supported by a grant (FOOD/2019/410-957) from the European

Union for the project “EU support to Livestock Disease Surveillance

Knowledge Integration” (LIDISKI).

I failed to change the section ' financial disclosure ' by The research was supported by a grant (FOOD/2019/410-957) from the European Union for the project “EU support to Livestock Disease Surveillance”.

The research was supported by a grant (FOOD/2019/410-957) from the European Union for the project “EU support to Livestock Disease Surveillance

Knowledge Integration” (LIDISKI)

5. Please state what role the funders took in the study. One of the request of the editor was to state the role the funders took in the study. We state that The funders had no role in study design, data collection and analysis, decision to publish, or preparation of the manuscript.

7. Please clarify whether this [conference proceeding or publication] was peer-reviewed and formally published. Another concern of the editor was the relation between the article submitted and the proceedings. previously published (https://link.springer.com/book/10.1007/978-3-031-53499-7. ). Our work has been peer reviewed and then included in the Proceedings. After the conference « Complex Network & their application 12 », we have been invited by the organizers to submit an extended version of our work to be part of the Plos Complex Sytems Special Collection. We followed the procedure suggested for the submission procedure that demanded to submit to the Plos One special issue complex network & their application. This work represents a further development of the work shown in the proceedings. As part of the process, we have been asked to add new material constituting 30% of the original manuscript (notably in methodology). This condition was respected. In particular, more details about the network and data collection have been added; more numerical experiments have been added to explore the space parameter and elucidate the relation between structure and the likelihood of being a sentinel node. The entire part relative to the backbone and the identification of sentinel nodes is a new and developed especially for this publication. We include a copy of the invitation at the end of the letter as well the submission guidelines provided. For more details, please contact the editors of PLOS Complex Systems(special.issuecna@gmail.com,Hocine Cherifi <hocine.cherifi@gmail.com>, Sabrina Gaito<gaito@di.unimi.it> ). 

8. In the online submission form, you indicated "access to data on request to Sandra Ijoma and Andrea Apolloni, article co-author". Concerning the availability of the data. Data have been collected in the framework of the Lidiski project, which funded the Ph.D. of Asma Mesdour and Sandra Ijoma. Dr Ijoma, co-author of this article, is submitting an article describing these and other mobility data. To preserve her work, the data will be made available at the moment of submission and under request until then. Nevertheless, a git repository (https://forgemia.inra.fr/umr-astre/manuscripts/mobility-and-ppr-diffusion-in-nigeria) has been created with anonymised version of data that can be used by reviewers and readers together with scripts used. More extended dataset will be available at the same repository following the submssion of the paper.

Reviewer #1: The study “Assessing the impact of structural modifications in the construction of surveillance network for transboundary animal diseases: the role of backbone and sentinel nodes” set out to identify critical districts in which to institute surveillance activities among other control activities of TADs with a special focus on Peste des petits ruminants (PPR). The study used data from 10 market surveys in 3 states and employed a mix of standard social network analysis and mathematical modelling approaches to identify the most probable “sentinel districts” in Nigeria. The authors present a strong case in regard to the urgent need to effectively control PPR and eventually eradicated through activities such as building strong and efficient surveillance systems, a key contribution to the global efforts to eradicate PPR.

However, the manuscript in current state suffers a few serious methodological and minor scholarly considerations that the authors should consider attending to so as to achieve their set objective (s).

General comments

The authors conveniently define ‘sentinel nodes’ as those areas that are rapidly infected first in case of disease introduction, a definition that leans more to anatomy (cancer research). 

Regarding the definition of "sentinel nodes," we understand the concern about its alignment with terminology traditionally used in anatomy, specifically in cancer research. We intend to use this term to convey the critical role these nodes play in the early detection and monitoring of disease spread within a network. In the context of our study, "sentinel nodes" are defined as districts (nodes in our network) that are often early infected in the event of disease introduction, thus serving as early alert points for potential outbreaks. This concept is crucial for designing and implementing an effective surveillance network, allowing timely intervention and control measures.

This article used a definition commonly applied in network literature to identify nodes for surveillance networks. For instance, Herrera et al. used "sensor nodes" to identify nodes likely to provide timely and accurate indications of epidemic activity. Similarly, Bajardi et al., Schirdewahn et al., and Bai et al. used the term "sentinel nodes" to refer to nodes that are often infected promptly (Schirdewahn et al., and Bai et al.) and provide information on the origin of the disease (Bajardi et al.). Our definition is thus grounded in established literature, particularly the works of Herrera et al., Bajardi et al., Schirdewahn et al., and Bai et al.

We added a paragraph to relate our definition of sentinel nodes to previous work, which we inspired to

“Previous works have attempted to identify nodes for surveillance networks. For instance, Herrera et al. used the term "sensor nodes" to identify nodes those nodes that are likely to provide timely and accurate indications of epidemic activity. Similarly, Bajardi et al., Schirdewahn et al., and Bai et al. used the term "sentinel nodes" to refer to nodes that are often infected promptly (Schirdewahn et al., and Bai et al.) and provide information on the origin of the disease (Bajardi et al.). “

 We clarified the definition of sentinel nodes in the text 

“ In this work, we defined sentinel nodes based on their vulnerability and the moment in time at which they are infected. There is no univocal definition of a node’s vulnerability. This article defines vulnerability as the probability of a node getting infected early in the epidemic. We used simulation results to identify sentinel nodes for each configuration and transmission probability. For each node, we estimated:

• We recorded for each simulation the time it has been infected (time of infection);

• We calculated the number of simulations times infection occurred before the epidemic peaked (frequency);

• We estimated the average frequency of infection and the standard deviation; 

• Based on the frequency distribution, the classical threshold (average frequency + 2 standard deviations) rule was used to classify nodes;”

References

Schirdewahn, F. et al. (2021) ‘Early warning of infectious disease outbreaks on cattle-transport networks’, PLOS ONE. Edited by S. Clegg, 16(1), p. e0244999. Available at: https://doi.org/10.1371/journal.pone.0244999.

Herrera-Diestra JL, Tildesley M, Shea K, Ferrari M. Network structure and disease risk for an endemic infectious disease [Internet]. arXiv; 2021 [cité 31 août 2022]. Disponible on: http://arxiv.org/abs/2107.06186

Bajardi P, Barrat A, Savini L, Colizza V. Optimizing surveillance for livestock disease spreading through animal movements. Journal of The Royal Society Interface. 7 nov 2012;9(76):2814 25. 

Bai, Y. et al. (2017) ‘Optimizing sentinel surveillance in temporal network epidemiology’, Scientific Reports, 7(1), p. 4804. Available at: https://doi.org/10.1038/s41598-017-03868-6.Bai, Y. et al. (2017) ‘Optimizing sentinel surveillance in temporal network epidemiology’, Scientific Reports, 7(1), p. 4804. Available at: https://doi.org/10.1038/s41598-017-03868-6.

 I would have thought that since the focus is control of TADs, a definition that is borrowed from epidemiology is more appropriate - sentinel nodes as to mimic sentinel monitoring or surveillance. These would be those districts intentionally identified to enable quick disease detection and patterns that guide more conclusive epidemiological investigations to determine spread mechanisms. 

An epidemiological approach involving intentional monitoring of districts to observe disease emergence/re-emergence over time and space could apply where logistical and financial resources permit routine data collection. However, in contexts with limited resources, such as many regions affected by TADs, implementing such comprehensive surveillance is often not feasible. In our study, we aimed to identify 

critical districts for surveillance activities based on their potential to be rapidly infected in the event of disease introduction: sentinel nodes are nodes where cases could be reported before the epidemic peak (because infected animals are introduced and/or naive residents' animals are infected). This aligns with the concept of sentinel nodes in complex network and in epidemiology, as these districts serve as early warning indicators of potential outbreaks. By focusing on these districts, we can effectively deploy surveillance and control measures to mitigate the spread of diseases like PPR. In our model, we didn’t delve into the details of the transmission at the local level while mainly focusing on the spread between nodes. Because of this choice, some dynamics that are considered necessary for local transition have not been considered. Nevertheless, this approach has been widely used in Bajardi et al., Schirdewahn et al., and Herrera et al. 

We explained our choices of modeling 

“The focus of this work is to study the propagation of PPR at the national level to identify possible locations where an outbreak could be reported and assess how the structure of livestock exchanges could impact the diffusion. To this end, we didn’t include livestock infection dynamics. We considered the inter-epidemic period, during which PPR is re-introduced in the area due to livestock movements from areas where cases are recorded. In the following, the term “Infected” nodes indicate nodes where new cases are registered due to the arrival of infected animals from other areas and where the transmission can occur to animals in the area. The main parameter ρ encompasses all the dynamics that could lead to the occurrence of at least a case in the area. We have omitted in this study. We have omitted several disease-related parameters that could impact the attack rate in the local population. This approach has been used in the past to identify candidate nodes for surveillance network “ 

For example districts that have a very high in-degree would be very good sentinels for disease surveillance because they simply receive animals from various origins that could potentially introduce disease. 

Surveillance of nodes with a high in-degree can be beneficial, and even nodes with high betweenness centrality can act as bridges or intermediaries in the network. However, the effectiveness of targeting high-degree or high-betweenness nodes may vary depending on the network structure and the characteristics of the disease. For instance, in the study by Colman et al.,(where the objective was to find out whether a strategy that distributes sentinel nodes in different regions of the network can be better than one that simply targets the highest degree nodes) highly modular or spatially integrated networks suggest that placing sentinels on nodes distributed across different region is preferable. Previous works by Herrera et al. have shown that indicators like eigenvector centrality are better suited to identify nodes that can be used as surveillance nodes, thus hinting at the importance of the position in the network. Furthermore, the timeliness is also related to the characteristics of the disease (in particular, the transmissibility). Because of this, in our work, we started with numerical simulations to identify sentinel nodes and, in a second step, identify the factors associated with the individual factors. 

Nevertheless, estimating centrality measures on incomplete data is not always easy. Hence, our study needs to modify our networks and determine which centrality is the most reliable. Our study shows that accounting only for observed network statistics in a partially documented connection framework could be misleading. Our findings suggest a potential role of peripheral nodes in disease transmission. The increase in the size of the network also changes the infection dynamics. Therefore, the identification of consistent set of sentinel nodes regarding network modification appears essential. 

Much as the authors apply robust techniques to identify the network backbone that has high potential for surveillance activities a much clearer approach for identifying sentinels for surveillance should have considered.

Our aim was to explore the potential utility of backbone nodes to identify sentinel candidates: we tested the hypothesis that since the backbone corresponds to an invariant set of nodes, nodes within the backbone could potentially serve as robust candidates for surveillance. To explore this hypothesis, we estimated how many sentinel nodes identified through our primary approach (nodes frequently affected before the peak) were also part of the network backbone. We found significant overlap between these sets of nodes, suggesting that nodes in the backbone may indeed be suitable candidates for surveillance. 

 For example, build the network and identify communities (if they exist) and each community selected districts with the highest selected centrality measure (s) – influential nodes and then those become sentinel nodes. 

As suggested in a previous point by Colman et al., for highly modular network this approach is a valuable one. However, this type of analysis is outside the scope of the present study but we are currently working on these questions.

Sentinel nodes would make much more sense if the authors then explored them over time (longitudinally) as cross-sectional view of the sentinels doesn’t tell a full picture. 

In our analysis, the network is considered static. One of the characteristics of “sentinel nodes” was the time of infection, which could vary depending on the characteristics of the disease, and the structure of the network. To identify the sentinel nodes we run 100 simulations for each combination of probability and network configuration and recorded if and when a node was infected before or after the peak. The distribution of the time of the infection, was used to identify sentinel nodes.

Unfortunately, we didn’t have the possibility to explore the sentinel nodes over a temporal network since no data on network evolution were available. This point was raised in the discussion. 

“As the quality of the prediction depends on the epidemic parameters, such as transmission and recovery rates, a more sophisticated model (SIR) would be capable of better capturing the interplay of animal movements (structure, volume, and temporal aspects) and transmission dynamic”

The authors need to explain more about their choice of approach to deal with missing/unreliable survey data especially inline with the natural network structure of livestock mobility in Nigeria

As in many sub-Saharan African countries, Nigeria doesn’t have a system to collect and map livestock movements, making it challenging to define a natural network. The work conducted by the Lidiksi project and presented in this article represents the first case where livestock mobility network has been reconstructed

---

## [Decision Letter · Decision Letter 1]

12 Sep 2024

Assessing the impact of structural modifications in the construction of surveillance network for Peste des petits ruminants disease in Nigeria: the role of backbone and sentinel nodes

PONE-D-24-11256R1

Dear Dr. MESDOUR,

We’re pleased to inform you that your manuscript has been judged scientifically suitable for publication and will be formally accepted for publication once it meets all outstanding technical requirements.

Kind regards,

Martin Chtolongo Simuunza, PhD

Academic Editor

PLOS ONE

Additional Editor Comments (optional):

Reviewers' comments:

Reviewer's Responses to Questions

**Comments to the Author**

1. If the authors have adequately addressed your comments raised in a previous round of review and you feel that this manuscript is now acceptable for publication, you may indicate that here to bypass the “Comments to the Author” section, enter your conflict of interest statement in the “Confidential to Editor” section, and submit your "Accept" recommendation.

Reviewer #1: All comments have been addressed

2. Is the manuscript technically sound, and do the data support the conclusions?

Reviewer #1: Yes

3. Has the statistical analysis been performed appropriately and rigorously? 

Reviewer #1: Yes

4. Have the authors made all data underlying the findings in their manuscript fully available?

Reviewer #1: Yes

5. Is the manuscript presented in an intelligible fashion and written in standard English?

Reviewer #1: No

6. Review Comments to the Author

Reviewer #1: I am generally satisfied with the responses to the queries I raised.

However, I believe that this manuscript would benefit from a dedicated "Conclusion" section summarising the takehome mesaage. The manuscript should still be revisited by the authors to rid it of some minor typographical errors that still exist in different sections.

7. PLOS authors have the option to publish the peer review history of their article (what does this mean?). If published, this will include your full peer review and any attached files.

Reviewer #1: **Yes: **Joseph Nkamwesiga

---

## [Editor Report · Acceptance letter]

20 Sep 2024

PONE-D-24-11256R1 

PLOS ONE

Dear Dr. MESDOUR, 

I'm pleased to inform you that your manuscript has been deemed suitable for publication in PLOS ONE. Congratulations! Your manuscript is now being handed over to our production team.

Kind regards, 

on behalf of

Dr. Martin Chtolongo Simuunza 

Academic Editor

PLOS ONE